# The Tumor Immune Microenvironment in Pancreatic Ductal Adenocarcinoma: Neither Hot nor Cold

**DOI:** 10.3390/cancers14174236

**Published:** 2022-08-31

**Authors:** Samuel J. S. Rubin, Raoul S. Sojwal, John Gubatan, Stephan Rogalla

**Affiliations:** Division of Gastroenterology and Hepatology, Department of Medicine, School of Medicine, Stanford University, Stanford, CA 94305, USA

**Keywords:** pancreatic ductal adenocarcinoma, tumor microenvironment, fibroblast, T cell, macrophage, neutrophil, myeloid-derived suppressor cell

## Abstract

**Simple Summary:**

In this review, we discuss the current understanding of pro- and anticancer immune responses in the tumor immune microenvironment of pancreatic ductal adenocarcinoma. We describe the duality and complexity of immune cell functions in the tumor microenvironment and also illustrate therapeutic approaches that modulate the antitumor immune response.

**Abstract:**

Pancreatic ductal adenocarcinoma (PDAC) is the most common pancreatic tumor and is associated with poor prognosis and treatment response. The tumor microenvironment (TME) is recognized as an important factor in metastatic progression across cancers. Despite extensive study of the TME in PDAC, the cellular and molecular signaling networks remain poorly understood, largely due to the tremendous heterogeneity across tumors. While earlier work characterized PDAC as an immunologically privileged tumor poorly recognized by the immune system, recent studies revealed the important and nuanced roles of immune cells in the pathogenesis of PDAC. Distinct lymphoid, myeloid, and stromal cell types in the TME exert opposing influences on PDAC tumor trajectory, suggesting a more complex organization than the classical “hot” versus “cold” tumor distinction. We review the pro- and antitumor immune processes found in PDAC and briefly discuss their leverage for the development of novel therapeutic approaches in the field.

## 1. Introduction

Pancreatic cancers include both exocrine and endocrine tumors. The most common type is pancreatic ductal carcinoma (PDAC), an exocrine tumor that accounts for approximately 90% of pancreatic cancers [1]. Other pancreatic exocrine tumors include pancreatic acinar carcinoma, which accounts for approximately 5% of pancreatic cancers, and several other rare tumors. Pancreatic endocrine tumors include rare neuroendocrine tumors (NETs), such as insulinomas, gastrinomas, and glucagonomas, some of which are associated with multiple endocrine neoplasia type 1 (MEN1) and other genetic syndromes. Herein, we focus on PDAC, as this is by far the most common and studied pancreatic tumor.

PDAC is increasing worldwide and is associated with high morbidity involving multiple organ system failures and high mortality [2]. Further, the tumors are particularly resistant to treatment relative to other primary origins, and 5-year survival remains under 10% despite decades of research and recent advances in immunotherapy, which have revolutionized the treatment and prognosis of other primary tumors. Multiple factors contribute to this poor prognosis. PDAC is often diagnosed at an advanced stage due to relatively asymptomatic progression through the early stages. Advanced imaging techniques are used to make an initial diagnosis with histopathology typically performed after surgery [1]. The tumors are characterized by marked fibrosis and hypervascularity in the tumor microenvironment (TME), which is further complicated by the tremendous heterogeneity of PDAC tumors across individuals. The TME is an amorphous entity consisting of the cells, molecules, and vessels and the associated spatial and signaling networks, all of which are important for malignant progression across cancer types. As the field of oncology has increasingly benefited from the perspectives of cancer immunology, great interest has developed in the immune microenvironment as a method to characterize and leverage the heterogeneity of PDAC in order to better understand clinical prognosis and design more effective therapeutic interventions.

Classical exposures such as tobacco and heavy alcohol use have been long associated with pancreatic cancer. Subsequent studies identified obesity and diabetes, increasingly common and recognized as inflammatory states, as independent risk factors for PDAC [3,4,5]. Genetic loci associated with other cancers and cancer syndromes, including *BRCA2* and *PALB2*, are also associated with PDAC. Approximately 90% of PDAC tumors exhibit Kirsten rat sarcoma virus (KRAS) failure, and many also demonstrate tumor protein (p53), mothers against decapentaplegic homolog 4 (SMAD4), and/or cyclin-dependent kinase inhibitor 2A (p16) mutations [6,7,8,9]. Nuclear factor kappa-light-chain-enhancer of activated B cells (NF-κB) dysregulation has also been implicated in both pro- and antitumor roles in the setting of PDAC. While much historical work on NF-κB suggested a primarily pro-tumor function of NF-κB activation in PDAC tied tightly to KRAS, more recent studies revealed simultaneous antitumor effects mediated by a complex signaling network affecting immune cells in the TME [10]. Activation of both canonical and noncanonical NF-κB pathways is well known to drive pancreatic injury in PDAC, and it is likely that modulation of the NF-κB pathway confers the recently observed protective effects via established control of immune cell function in the TME [11].

Nevertheless, PDAC is still considered to possess a relatively low mutational load compared to other tumors [12], which typically leads to low levels of immune cell infiltration [13]. Previously, this observation led to the characterization of PDAC as an immunologically privileged tumor, which is referred to as “cold” relative to other tumors with high mutational burdens and thus better immune recognition known as “hot” tumors. However, there is increasing recognition that immune cells, whether infiltrating or resident, play significant roles in PDAC pathogenesis. Thus, PDAC may not adhere to the traditional “hot” versus “cold” tumor distinction, but rather exists on an immune temperature spectrum that includes both “hot” and “cold” zones in the TME. In this review, we focus on the tumor immune microenvironment. Herein, we use TME and tumor immune microenvironment interchangeably since it is not possible to distinguish immune elements from the rest of the TME. We synthesize the multifaceted roles of distinct lymphoid, myeloid, and stromal cell types in the tumor immune microenvironment of PDAC and briefly discuss ongoing work to understand this complex network in the treatment of PDAC.

## 2. The PDAC Tumor Immune Microenvironment

The tumor immune microenvironment is immensely heterogeneous in PDAC, which is largely responsible for its extensive study and lack of mechanistic clarity to date [14]. Key elements of its heterogeneity stem from cancer-associated fibroblasts (CAFs), endothelial cells, and numerous subsets of immune cells (Figure 1). Fibroblasts and other PDAC stromal cells largely promote tumor evasion from T-cell control. In a large multiomic study, sub-tumor microenvironments (subTMEs) correlated with clinical heterogeneity and were primarily defined by fibroblast plasticity [15]. “Hot” subTME regions were characterized by activated CAFs that promoted the growth of undifferentiated tumor cells, whereas “cold” subTME regions contained less activated CAFs and tumor suppressive features. Other studies that focused on soluble mediators of the TME found that interleukin-1β (IL-1β) secreted by the tumor stroma in PDAC promotes polarization of macrophages to the M2 phenotype, and proliferation of myeloid-derived suppressor cells (MDSCs), regulatory B cells (Bregs), and T-helper 17 (Th17) cells, which promote tumorigenesis [16,17]. Blockade of IL-1β in a mouse model of PDAC improved tumor-infiltrating lymphocyte (TIL) numbers and CD8^+^ T-cell responses, which inhibit tumorigenesis. 

Thus, these stromal cells are one way that PDAC defies the “hot” and “cold” immune response dichotomy. Large transcriptomic studies suggest important roles for numerous immune cell subsets in the TME [18], which we discuss by lineage compartment.

## 3. Lymphoid Compartment

Lymphoid cells include B cells and T cells, the latter of which have been more studied in PDAC. Previously, T cells were considered uniformly tumorigenic in PDAC, owing to their “cold” tumor designation. However, it is increasingly recognized that T cells play dual roles to permit and, when activated, suppress tumor progression in PDAC. A recent large multiomic study of PDAC demonstrated the complexity of suppressive interactions that prevent T cells from controlling the tumor growth [19]. Tumor-infiltrating lymphocytes (TILs) are a subset of CD8^+^ T cells that generally inhibit tumor growth by cytotoxicity and other mechanisms. Dysfunctional TILs express a heterogeneous array of immune checkpoint receptors in PDAC, reflecting an exhausted phenotype. In particular, TILs upregulated the T-cell immunoreceptor with Ig and ITIM domains (TIGIT), an inhibitory checkpoint receptor that interacts with dendritic cells (DCs) and macrophages. These DCs and macrophages then express anti-inflammatory interleukin-19 (IL-10), which interferes with the ability of cytotoxic T cells to mitigate PDAC. TIGIT also mediates TIL function in other cancers, and clinical trials are ongoing to evaluate anti-TIGIT immunotherapies in combination with other checkpoint inhibitors.

Soluble factors also contribute to TIL dysfunction in PDAC. C-X-C motif chemokine ligand 1 (CXCL1) promotes a dominant noninflamed T-cell phenotype, and ablation of the chemokine promoted CD8^+^ T-cell infiltration and immunotherapy response in a mouse model of PDAC [20]. The anti-inflammatory cytokine transforming growth factor-β (TGF-β) also promotes tumor progression in PDAC by inhibiting of immune activation, and deletion of the TGF-β receptor in CD8^+^ T cells enhanced infiltration and cytotoxicity of TILs in mouse models [21]. Moreover, CD4^+^ T-helper cells also play opposing roles in the TME of PDAC tumors [22]. T-helper 1 (Th1) and T-helper 2 (Th2) cells induced in the TME by proinflammatory cytokines enhance TIL antitumor activity, while the role of Th17 cells remains poorly understood but likely promotes tumor growth depending on the role of interleukin-17 (IL-17). Regulatory T cells (Tregs) also supply TGF-β ligands to fibroblasts, inhibiting the growth of the tumor [23]. Furthermore, there was increased differentiation of inflammatory fibroblasts, accelerated tumor progression, and enhanced C-C chemokine receptor type 1 (CCR1)-mediated MDSC infiltration when Tregs were depleted in a PDAC mouse model, suggesting that Tregs inhibit tumorigenesis in PDAC. This observation strikingly contrasts with other cancers such as melanoma where Treg depletion significantly enhances tumor control by CD8^+^ T cells. In addition, infiltrating T follicular helper (T_FH_) cells produced C-X-C motif chemokine ligand 13 (CXCL13) and interleukin-21 (IL-21), which improved CD8^+^ T and B cell infiltration, activation, and maturation that could be inhibited by programmed cell death protein 1 (PD-1) signaling, demonstrating additional antitumor T-cell functions [24]. Thus, distinct T-cell subsets play dual roles in both inhibiting and promoting tumorigenesis in PDAC.

Natural killer cells (NK cells) such as lymphocytes from the innate immune system have not been profoundly studied for their role in PDAC TME and thus their role remains unclear [25]. More recently, NK cells have been investigated and it was shown that certain subsets of NK cells, CD56^dim^ and CD16^neg^, were reduced in the bloodstream yet enriched in TIL of PDAC, which can be seen as a sign that NK cells may suppress tumor progression [26]. However, more research needs to be done to understand NK cells’ role and potency for immune therapies.

## 4. Myeloid Compartment

Macrophage polarization has a well-established influence on the TME across multiple primary cancers. Immunosuppressive M2-like tumor-associated macrophages (TAMs) are associated with poor prognosis by their overall density in the tumor, albeit improved prognosis when localized in close proximity to PDAC tumor cells [27,28]. This duality may reflect the effect of other nearby cells, which ongoing multidimensional spatial studies aim to determine. Hypoxia-inducible factor-2α (HIF2α) production by fibroblasts in the relatively hypoxic stroma of PDAC also promotes M2 macrophage polarization, as evidenced by a recent mouse study [29]. In contrast, macrophages also express colony stimulating factor 1 receptor (CSF-1R) and focal adhesion kinase (FAK) that mediate signaling cascades to promote MDSC, TAM, and Treg infiltration and are under investigation as therapeutic targets for the treatment of PDAC [30]. In particular, MDSCs interfere with T-cell activation and infiltration. In a mouse model of PDAC, the histone deacetylase inhibitor entinostat was used to reprogram MDSCs such that when combined with anti-PD-1 or anticytotoxic T-lymphocyte-associated protein 4 (CTLA-4) immunotherapies, TILs were more activated and outcomes were improved with a combination therapy [31]. Overall, M2 macrophages, M2-like TAMs, and MDSCs play both “hot” and “cold” roles in the PDAC TME.

Other myeloid lineage cells have also been studied in PDAC. A deficiency in conventional DCs (cDCs) led to tumorigenic Th17 cell responses and tumor progression despite the presence of neoantigens [32]. In this mouse study, increased cDC production in early PDAC led to tumor progression, while increased cDC production in advanced PDAC promoted immune control of the tumor and response to radiation therapy. In addition, distinct DC subsets at secondary sites have been shown to promote PDAC metastasis via modulation of CD8^+^ T cells [33], again representing the duality of immune cell functions in PDAC.

Neutrophils are also abundant in the PDAC TME and are associated with poor clinical prognosis. Mouse models of PDAC treated with lorlatinib (an FDA-approved inhibitor of anaplastic lymphoma kinase, or ALK) demonstrated a reduction in tumor progression mediated by inhibition of neutrophil development and mobilization [34]. Treatment with lorlatinib improved response to anti-PD-1 immunotherapy and promoted activation of CD8^+^ T cells. Mast cells have also been shown to produce angiogenic factors that promote PDAC tumor growth and metastatic progression [35]. Thus, myeloid lineage cells play numerous, nuanced roles in promoting, and to a lesser extent modulating, PDAC tumorigenesis.

## 5. Role of Tumor Immune Microenvironment in Treatment

Evidently, distinct immune cell subsets play multiple, opposing roles in the PDAC TME. The cornerstone of current PDAC therapy remains a combination of chemotherapy, radiation, and surgery, which will likely continue to play a role in addition to next-generation therapies [1]. Immune checkpoint inhibitor immunotherapy that revolutionized the field of oncology over the last decade has yet to make the same monumental advancement in PDAC given its poor therapeutic response. Given its growing prevalence and morbidity and mortality, considerable research is dedicated to the development of next-generation immunotherapies for PDAC and other treatment-resistant tumors (Figure 2). In targeting T cells, the solution may stem from combining FDA-approved therapeutics, such as anti-PD-1 and anti-CTLA-4 checkpoint inhibitors with little activity in PDAC, with novel checkpoint targets, such as TIGIT [19] and tumor necrosis factor receptor superfamily member 4 (OX40) [36]. Stimulator of interferon genes (STING) and Toll-like receptor (TLR) agonists are also in clinical development to promote DC activation and thereby improve T-cell cytotoxicity. Other programs focus on modulation of MDSCs and M2 macrophages. Alternative platforms are being used to target these elusive cell populations, including exosomes containing targeted siRNAs for reprogramming M2-TAMs to inflammatory M1 phenotypes and recruiting cytotoxic T cells [37]. In addition, mRNA technology is being leveraged for the development of cancer vaccines to treat PDAC using antigens defined by large screening studies [38], and still, other programs are evaluating the utility chimeric antigen receptor (CAR) T, oncolytic virus, vaccine, small molecule, and antibody technologies.

In addition, immune cells can also act as prognostic markers. A recent meta-analysis by McGuigan et. al. found several promising immune cells as biomarkers for disease progression [40]. The authors found evidence that invasion of CD4 and CD8 positive T cells were associated with improved disease-free survival, whereas CD163 positive cells were associated with reduced overall survival [40]. Moreover, several studies have shown αvβ6, a member of the integrin family, is overexpressed in pancreatic cancer, which suggests this as another avenue for diagnostic and therapeutic innovation [41,42].

## 6. Conclusions

While the role of fibrosis in the PDAC TME dominates much of the classical literature in the field, it is clear from recent and ongoing research that immune cell subsets play nuanced and pathologically significant roles in tumorigenesis. Primarily, M2-polarized macrophages promote PDAC tumorigenesis along with neutrophils and exhausted TILs that permit tumor growth. On the other hand, certain DC subsets and T_FH_ cells activate cytotoxic T cells to inhibit tumor progression. The roles of these cells are nuanced and, as such, do not fit into the traditional “hot” or “cold” dichotomy. As the availability of suitable clinical samples is always a limiting factor in translational studies and mouse models are never perfect, organoids are also being explored as tools to advance the field. A recent study demonstrated that primary human organoids reflected the PDAC stromal and immune compartments, including TILs, cancer-associated fibroblasts, and the three-dimensional TME, which will help facilitate ongoing research efforts [43]. Next-generation therapies will no doubt be multifaceted and likely multiagent, involving synergistic approaches aimed at modulating the immunosuppressive TME and enhancing the cytotoxic immune response.

## Figures and Tables

**Figure 1 cancers-14-04236-f001:**
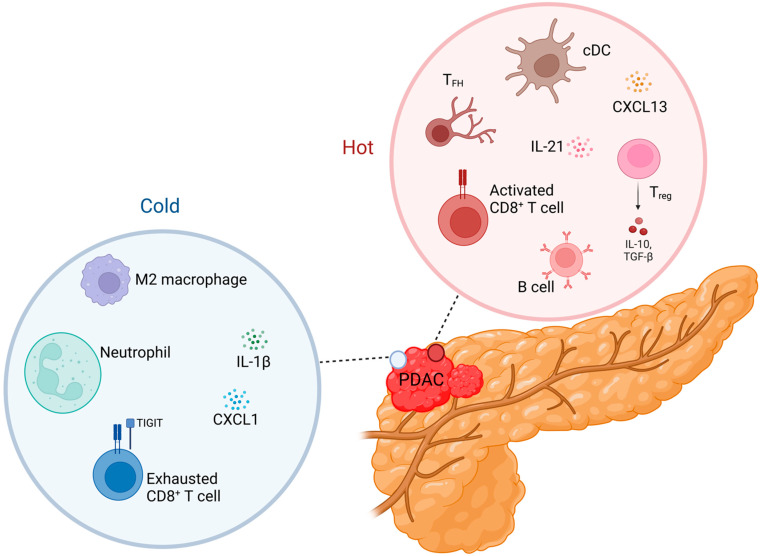
The tumor immune microenvironment in pancreatic ductal adenocarcinoma involves complex, opposing interaction networks. Presence of both “hot” (red circle) and “cold” (blue circle) microenvironments defies the classical paradigm in which these states are mutually exclusive. Instead, processes characteristic of each state are found within the same tumor. IL-1β, interleukin-1β; CXCL1, C-X-C motif chemokine ligand 1; TIGIT, T-cell immunoreceptor with Ig and ITIM domains; T_FH_, T follicular helper cell; IL-21, interleukin-21; cDC, conventional dendritic cell; CXCL13, C-X-C motif chemokine ligand 13; Treg, Regulatory T cell; IL-10, interleukin-10; TGF-β, transforming growth factor-β. Created with BioRender.

**Figure 2 cancers-14-04236-f002:**
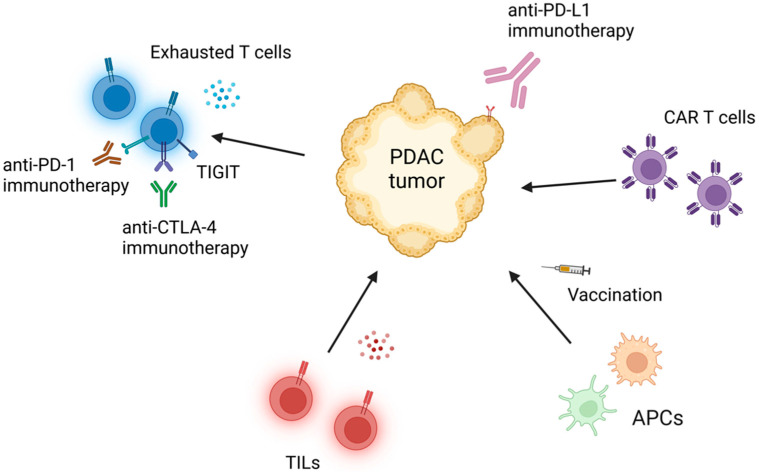
Immunotherapeutic approaches to the treatment of PDAC that leverage the tumor immune microenvironment. APC, antigen presenting cell; CAR, chimeric antigen receptor; CTL, cytotoxic T cell; LAG-3, lymphocyte-activation gene-3; PD-1, programmed death receptor-1; PD-L1, programmed death-ligand 1; T_ex_, exhausted T cell. Adapted from [39]. Created with BioRender.

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
