# Peer review of "The Tumor Immune Microenvironment in Pancreatic Ductal Adenocarcinoma: Neither Hot nor Cold"

_cancers, 2022, doi:10.3390/cancers14174236_

Round 1
Reviewer 1 Report
This review manuscript by Samuel J. S. Rubin, and colleagues comprehensively summarized current understanding of pro- and anti-tumor immune processes found in Pancreatic ductal adenocarcinoma (PDAC). They discussed immune cells and associated signaling pathways potential for the development of future therapeutics for the PDAC treatment. The authors summarized the function of immune cells in the development and inhibition of carcinoma. Lastly, the authors discussed the role of inhibitory mechanism in the treatment of PDAC. Also, this review proposed that development of new immune-mediated strategies for the treatment of PDAC.
However, here are the concern below with current version of the manuscript.
The review should also summarize the function of other CD4+ T helper cells, such as Th1, Th2, and Th17 cells in PDAC context, such as: Xiaomeng Liu et. al. Mol Cancer. 2019; 18: 184. “The reciprocal regulation between host tissue and immune cells in pancreatic ductal adenocarcinoma: new insights and therapeutic implications” The authors need to discuss the related literatures.
In addition, there are several places’ where references are missing.
Author Response
We thank the reviewer for the positive suggestions to improve our manuscript and edited it accordingly!
Reviewer 2 Report
This is a well-written brief review on the tumor immune microenvironment in pancreatic ductal adenocarcinoma. Some minor suggestions are described as follows:
1. Full names of some abbreviations, such as MEN1 and TIGIT, should be provided.
2. The regulation and distribution of the immune temperature spectrum in TEM could be discussed elaborately.
Author Response
We thank the reviewer for the positive suggestions to improve our manuscript and edited it accordingly! In addition, we went over the language again despite most authors being native English speakers.
Reviewer 3 Report
The authors present a very insightful and interesting, well-written review on the immune microenvironment in pancreatic ductal carcinoma. I strongly recommend the publication of this review after minor modifications:
1-The role of NF-kappaB signaling patwhay, a key pathway in immune-oncology, it is not mentioned.
Authors should mention the increasing role of NF-kappaB in the control of the immune microenvironment in PDAC. At least, excellent reviews should be mentioned in the introduction:
Kabacaoglu, D., Ruess, D. A., Ai, J. & Algül, H. NF-κB/Rel Transcription Factors in Pancreatic Cancer: Focusing on RelA, c-Rel, and RelB. Cancers 11, 937 (2019)
Silke, J. & O’Reilly, L. A. NF-κB and Pancreatic Cancer; Chapter and Verse. Cancers 13, 4510 (2021)
2-In section 5, last paragraph, "However" should be replaced by "In addition" as follows: "In addition, immune cells can as well act as prognostic markers..."
3-The resolution quality of Figure 1 is not acceptable. The authors should provide with a much better quality image and the words "hot" and "cold" should be written next to the corresponding circles.
Author Response
We thank the reviewer for the positive suggestions to improve our manuscript and edited it accordingly! The references suggestion were excellent and were implemented. Thank you.